# The Use of Natural Collagen Obtained from Fish Waste in Hair Styling and Care

**DOI:** 10.3390/polym14040749

**Published:** 2022-02-15

**Authors:** Joanna Igielska-Kalwat, Ewa Kilian-Pięta, Sława Połoczańska-Godek

**Affiliations:** 1Faculty of Cosmetology, University of Education and Therapy, Grabowa 22, 61-473 Poznań, Poland; s.godek@wseit.edu.pl; 2Symbiosis Laboratory, Poznań Science and Technology Park, Rubież 46H, 61-612 Poznań, Poland; ewa.kilian@symbiosis.pl

**Keywords:** biodegradability, collagen, bio-based polymers, TrichoScope Polarizer Dino-Lite (MEDL4HM), scanning electron microscope (SEM), cosmetics formulations

## Abstract

Chemically speaking, polymers are multi-molecular compounds that have specific physicochemical properties. Hair cosmetics utilize their ability to create a protective film and make the cosmetic formulation more viscous, which facilitates its application. Natural polymers are encountered in nature, but, in hair cosmetics, artificially modified ones are more often used. Unfortunately, artificially modified polymers are characterized by high resistance to biological factors, which creates an ecological problem. Another reason for a search for natural polymers is their milder action when compared to synthetic ones. One of the new sources of obtaining collagen is the waste connective tissue materials of aquatic animals—skins, spines, dorsal chords and scales, and swim bladders. These raw materials are most often disposed of in landfills, processed into fish meal, or destined for food for animals. The conducted research was aimed at proving the action of natural collagen in hair cosmetics as a substitute for synthetic polymers. In the patients using collagen laminate, it is possible to notice the complete elimination of excessive sebum production, restoration of the correct pH value, and reduction in skin inflammations.

## 1. Introduction

Polymers can be found in all care and modeling preparations, such as gels, foams, and sprays. This is due to their natural features; the compounds leave a characteristic film on the hair, one that is difficult to remove during ordinary head-washing. These compounds add volume to hair strands without causing them to clump together and make them easier to comb. Conditioners use positively charged compounds that perfectly penetrate negatively charged damaged hair. The strands covered with a delicate film become shiny and look healthier. The film formed by polymers protects the hair very well against UV radiation and high temperature. The search for new synthetic polymer alternatives has started mainly due to the fact that, despite some of the positive properties of synthetic polymers, they still accumulate on the hair shaft. Their filling properties create a protective layer on the hair or skin, which, however, clogs the pores and makes it difficult for the skin to breathe, acting as a blocker. In the case of hair, it creates a tight, aggravating film through which no beneficial active substances can penetrate [1]. Long-term accumulation on hair may cause it to break down completely [2]. The leveling and smoothing effect of polymers is deceptive because it is only an optical impression, lasting during the use of the cosmetic that contains synthetic polymers [3]. In order to remove the tight film formed by synthetic polymers, stronger detergents should be used. This can destroy the natural lipid coat of the scalp and weaken and dry the hair. Another negative feature of synthetic polymers is, as mentioned earlier, their excessive accumulation on the surface. The hairstyle becomes heavy and not amenable to styling [4,5,6,7,8]. The use of stylers with such polymers can also affect the coloring process since dye pigments cannot reach deep into the hair with ease. The result is a shade of hair that differs significantly from the intended color. It is also worth noting that the discussed polymers are not subject to biodegradation processes. As already mentioned in the research, the introduction of alternative products was undertaken for synthetically obtained polymers, which were used as products for styling and hair care. These compounds were replaced with collagen obtained from fish waste.

Collagen is a building block of skin, bones, and connective tissue membranes. It makes up about 30 percent of the protein contained in mammals. Thanks to its unique physicochemical properties, it is widely used in industry, including the pharmaceutical and cosmetic industries.

Fish processing for consumption generates significant amounts of various types of waste materials, which are generally used as an ingredient of animal feed. There are indications of the possibility of a more rational use of some of them for market products intended for humans and not only for fodder.

Until now, mammalian collagen has been used in the world of cosmetology. However, as a result of the Creutzfeldt–Jakob disease and BSE epidemic, this collagen has been systematically replaced with a preparation obtained from fish with much better cosmetic properties, which is related to the use of the so-called acid hydration, which allows the structure of the triple helix to be preserved. This allows for the preservation of the biological activity of collagen, which, by penetrating the extracellular space of the epidermis, influences the activity of keratocytes and fibroblasts, resulting in a number of cosmetic and dermatological effects. Moreover, the solubility of collagen present in the skin of fish allows obtaining the native form of collagen in the final product—collagen hydrogel—using an extremely delicate method of obtaining this protein under conservative conditions [8,9]. The obtained collagen is used both in home hair care and also as an active substance constituting the basis for professional treatments, such as “Botox”, lamination, and collagen hair straightening. According to statistics, hair care products are the most frequently chosen care products in recent years.

The aim of this study was both to create cosmetic formulations used for hair styling, washing, and conditioning based on natural polymers and to compare the results of probands using synthetic organosilicon and natural polymers. The main active substances were collagen obtained from fish and hyaluronic acid [9].

## 2. Materials and Methods

### 2.1. Materials

The research used collagen derived from fish skin (FS collagen—fish skin collagen) obtained by acidic hydratation method patented by Przybylski [10]. It consists of hydration of the cleansed skin in a solution of lactic acid at a concentration of 0.1 to 1.5%. The hydration process is carried out in glass containers at the temperature ranging from 15 to 20 degrees Celsius for a period of 24 to 48 h. As a result of multiple filtrations with the use of natural silk of increasing density, cell elements, pigments, and the remains of an acid solution were separated from pure collagen. Natural silk filters have a structure similar to that of collagen, allowing the preservation of its native structure [11].

The oil phase was prepared in a 100 mL beaker (basis weight and names of substances are given in Table 1). This phase was melted in a water bath at 80–85 °C. After cooling the mixture, collagen was added and hyaluronic acid was homogenized until a homogeneous emulsion was obtained using IKA T 50 basic Ultra-Turrax homogenizer with rotational speed of 6000 rpm [12]. All was then combined using a magnetic stirrer, mixing vigorously. The mixture was left in the stirrer to cool completely. A gray homogeneous emulsion was obtained (Figure 1A,B).

The remaining active substances presented in Table 1 were added to the collagen obtained in the above manner.

It is very important to test the pH value of the product, which is characterized by the value of pH = 5.50 [±0.3].

### 2.2. Tests Confirming the Safety and the Manufacturer’s Declaration for the Collagen Laminate

In the subsequent part of the study, tests were performed in accordance with the Act of 4 October 2018 on Cosmetic Products (Polish Journal of Laws of 2018, item 2227) (Art. 11(3) [13]. Their aim was to assess the impact of the cosmetic on the safety of human health—including studies on skin tolerance to the tested cosmetic at the place of its application. Apparatus confirmation of the type of actions of the preparations declared by the manufacturer was carried out. The research was conducted following the approval of the Bioethics Committee, resolution number 640/20. Tests confirming or excluding the properties and performance of collagen, a natural polymer, declared by the Ordering Party, were carried out at probands’ homes and in the Dermatology Center (Centrum Dermatologii Sp. Z o.o.) in Poznań under the supervision of a dermatologist, chemist, and cosmetologist. The probands were volunteers. A group of 30 people was selected for the study, corresponding to the indications for the use of the product submitted for experiments. The selection of probands was carried out according to the preliminary assessment of the condition of their hair, scalp, and health. Due to the intended use of the product, the study involved people who declared their willingness to improve the appearance and condition of the scalp and hair [14].

Methodology of the research: at home (home panel).

The test probands qualified for the research received one package of collagen laminate. They undertook to use the received sample on a regular basis in accordance with the application rules suggested by the manufacturer. Probands should not have been using preparations of the same or analogous purpose during the time of the study. Volunteers were to observe the reaction of the scalp and hair at the application sites. They should note in detail the remarks concerning the functional properties of the tested cosmetic. In the event of any negative symptoms at the place of application of cosmetics, they were to stop using cosmetics and contact the person conducting the examination immediately. No special requirements were imposed on the participants of the study, assuming that the effects of this type of cosmetic should be tested in the natural conditions in which it will be used in practice. The results of the experiment could only be influenced by factors such as health, type and condition of hair and skin, genetic conditions, individual traits, individual preferences, lifestyle, environmental conditions, etc. A group of 30 people who used cosmetics containing synthetic organosilicon polymers for conditioning, washing, and styling their hair was also studied.

### 2.3. Apparatus Tests Carried Out with the Help of TrichoScope Polarizer Dino-Lite (MEDL4HM)

The research used the imaging method involving a trichological microcamera with the assessment of the condition of the skin and the follicular area. TrichoScope Polarizer Dino-Lite (MEDL4HM) is a device designed for the examination of the scalp and hair analysis. With a very high magnification, up to 200 times, a single hair can be viewed in great detail. The built-in polarization filter minimizes the effect of glare on the scalp or hair [15]. As already mentioned, the collagen laminate study was conducted in cooperation with the Symbiosis Dermatology Center on a group of 30 probands. Before starting the test, each volunteer was assigned a cosmetic sample. For the subsequent 100 days, the study participants were asked to test the samples at home (in-home-use-test methodology). With the help of a microcamera, the experimenter examined the condition of the scalp area and hair of the patients before and after using the preparations. A blank test was performed prior to starting product application. The second test was carried out after 100 days, during which the probands undertook to use the preparation regularly.

### 2.4. Apparatus Test Performed Using the SEM(Scanning Electron Microscope) Method

During the blank test, hair was collected from the probands’ both temples, the center of the head, and the occiput in order to perform imaging studies using a scanning electron microscope. SEM is a kind of electron microscope enabling the observation of the topography of the tested material (hair). It is used to observe materials on a nanometric to micrometric scale. With scanning microscopy, it is possible to precisely establish the hair morphology, the condition of the hair bulb and stem, the type and size of damage to the hair tissue, as well as to identify factors affecting the hair to determine its structural changes or to analyze its chemical composition. After a hundred days, the test was repeated. Subsequent samples were taken from all probands, and imaging was performed using SEM. All participants remained in the study until the end [16].

## 3. Results and Discussion

### 3.1. Instrument Tests Carried Out with the TrichoScope Polarizer Dino-Lite (MEDL4HM)

Probands using collagen laminate and preparations containing synthetic organosilicon polymers were tested during the blank test and after 100 days. The research used the imaging method involving a trophological micro camera with the assessment of the condition of the skin and the follicular area (TrichoScope Polarizer Dino-Lite (MEDL4HM)). A compilation of photos presented below shows the difference before the application of laminate collagen and after the 100-day study. In the most severe cases, the frequency of follow-up examinations was increased. The Figure 2, Figure 3, Figure 4 and Figure 5 below were taken of different areas of the scalp. In addition, people who applied synthetic organosilicon polymers to the scalp and hair were examined as a control group.

PATIENT A and B (blank determination)—a group of patients who started using natural polymers in the form of a collagen laminate product.

PATIENT V and X (blank determination)—a group of patients who used and continued the application of preparations containing synthetic organosilicon polymers.

The above patients had symptoms of peeling epidermis. They constitute examples of wrong hair care as they used synthetic organosilicon polymers with siloxane structure in which all silicon atoms are linked to alkyl or aryl groups. These compounds had a negative effect—mainly on the scalp—and increased its pH value. An alkaline pH value leads to the development of many dermatoses. As can be seen from the measurements of both zero groups, along with the application time of the synthetic silicones mentioned above, the probands struggled with seborrheic dermatitis. In some of the probands, the lesions turned into irregular erythema spots with yellow scabs. There were also diffuse inflammatory lesions covering the entire surface of the scalp [17]. The dried layer of dead skin and sebum formed a crust. Moreover, the patients complained of itchy wounds, which they often scratched. The hair was also thinned.

PATIENT A and B (test after 100 days)—a group of patients who used natural polymers in the form of a collagen laminate product.

PATIENT V and X (test after 100 days)—a group of patients who used preparations containing synthetic polymers.

In the patients using collagen laminate, it is possible to notice the complete elimination of excessive sebum production, restoration of the correct pH value, and reduction in skin inflammations. Based on the research, it can be seen that the active substances used confirm the effects declared by the manufacturer. After just one application of the laminate, the scalp was cleansed [18].

The substances used in the product are responsible for this effect. One of the few forms of collagen that is not subject to chemical treatment in cosmetology and trichology is type III collagen—interchangeably called native collagen or tropocollagen [19].

Type III collagen is obtained most often from the skins of young animals. Recently, the consumption of marine collagen, isolated from fish processing waste, such as skin, bones, fins, scales, etc., has been growing. Using waste instead of throwing it away allows the maximum use of resources, and it is undoubtedly an environmentally friendly approach [20]. The source of marine collagen created by our laboratory was fish. It is also not without significance that collagen obtained from marine resources is characterized by biocompatibility and a low probability of sensitization; plus, it is soluble in water, biodegradable, and easy to obtain [21].

It is these features of collagen obtained from sea water resources that made it the main ingredient of the product. The formulation uses collagen derived from fish skins, which plays a role analogous to that of tropocollagen. In addition, it is characterized by a low molecular weight, thanks to which it effectively soothes irritations and supports skin regeneration [22,23,24]. Unfortunately, after 100 days of the study, the control group that used only formulations with synthetic organosilicon polymers did not achieve any improvement. As already mentioned, these polymers create a strong occlusive film that does not allow the scalp to breathe or active substances that could improve the skin condition of the probands to penetrate deeper.

### 3.2. Apparatus Test Performed Using the SEM (Scanning Electron Microscope) Method

On the basis of research carried out with the use of a scanning electron microscope (SEM) that enables the observation of the topography of the tested hair, differences in its condition can be noticed (blank sample and tests after 100 days). A compilation of photos presented below shows the difference before the application of the laminate collagen and after the 100-day study. In the most severe cases, the frequency of follow-up examinations was increased. The photos below were taken of different areas of the scalp. In addition, people who applied synthetic organosilicon polymers to the scalp and hair were examined as a control group.

PATIENT A and B (blank determination)—a group of patients who started using natural polymers in the form of a collagen laminate product (Figure 6).

PATIENT V and X (blank determination)—a group of patients who used and continued the application with preparations containing synthetic organosilicon polymers (Figure 7).

The SEM analysis of the hair structure using a scanning electron microscope confirmed its layered structure. Photos 5 and 6 show the outer layer of the hair—the cuticle. The visible torn, cracked, and open hair cuticles testify to a high degree of damage. As can be seen in the photo showing the hair before using the preparation, it was damaged and characterized by hair sheath of open cuticles. A casing composed of open cuticles gradually ceases to fulfill its protective function. The cuticles open, among others, under the influence of alkaline pH and heat, resulting in high insolation, mechanical treatments, and poor styling with the use of aggressive chemicals. The participants of the study subjected their hair to mechanical and thermal treatments as well as the use of chemicals. Open cuticles result in the hair losing water faster and becoming dry, dull, rough, and more sensitive to external factors [25,26,27].

PATIENT A and B (trial after 100 days)—a group of patients who used natural polymers in the form of a collagen laminate product (Figure 8).

It can be said that the mechanism of action of collagen with a low degree of hydrolysis (M = 125,000) consists in the production of a protective film that closes the exposed cuticles and the cortex of the hair. It was found that further application would result in an increased protection through the closure of hair cuticles, i.e., improvement of their condition. In addition to the collagen used, the essential ingredients used in the product are biomimetric ceramides synthesized by the participants of the study [28]. They were obtained in a green chemistry process without the use of solvents from two saturated vegetable fatty acids. These substances strengthen the hair fibers, smoothen the hair surface, improve resistance, and give shine. Biomimetic ceramides are combined with avocado and grape oil. They contribute to the strengthening of the hair structure.

PATIENT V and X (trial after 100 days)—a group of patients who used preparations containing synthetic polymers (Figure 9).

Figure 9 presents probands who continued to use synthetic organosilicon polymers in whom no major change of the hair structure was noticed. The excessive application of synthetic polymers resulted in the accumulation of this substance, the tilting of the hair cuticle, and, in the final stage, complete tearing.

## 4. Conclusions

Our hair is exposed to numerous negative factors (Figure 10—Activoil Kerox Pro Release; Innovacos Corp: Mount Arlington, NJ, USA, 2018–2020).

Knowledge of the biochemical structure of collagen follows the knowledge of the amino acid composition. The collagen triple helix is made of three tropocollagen chains connected with each other by means of covalent-technical bonds by introduction into the middle-glycine. The spiral structure can be achieved with alanine, glutamine, histidine, leucine, methionine, tyrosine, and tryptophan. Valine and isoleucine, due to the large size of the chains, cannot participate in the formation of the structure of the helix generator. Serine, threonine, proline, and hydroxyproline provide good helix structures. The first two groups from the linkage of hydrogen bonds are formed by their hydroxyl ones [17,18,19,20]. In the case of proline, the atomic atoms are in a heterocyclic compound, excluding the possibility of rotation around the carbon–nitrogen bond and formation of hydrogen bonds inside. As a first proposition of proline, the chain may bend or even form a loop. Thanks to the use of natural biodegradable collagen combined with hyaluronic acid, ceramides, and raspberry oil, an alternative has been created to the controversial synthetic organosilicon polymers. This alternative also creates an occlusive film preventing against the negative effects of solar radiation, temperature changes, pollution, pathogens, and aggressive products used for hair styling [29,30,31,32]. In addition, the product is obtained from natural waste, which protects our environment. Collagen laminate is a universal product that can be used for washing, conditioning, and styling hair (Figure 11—Activoil Kerox Pro Release; Innovacos Corp: Mount Arlington, NJ, USA, 2018–2020). The developed method allows to obtain collagen in a native form in the final product. The collagen obtained in this way can be used not only in the form of a gel or solution but also in a solid form, replacing the bovine collagen used so far. The obtained collagen was used in the form of a gel as a hair care product. Based on the assessment of the situation, the condition of the hair and skin can be improved 100%.

**Figure 10 polymers-14-00749-f010:**
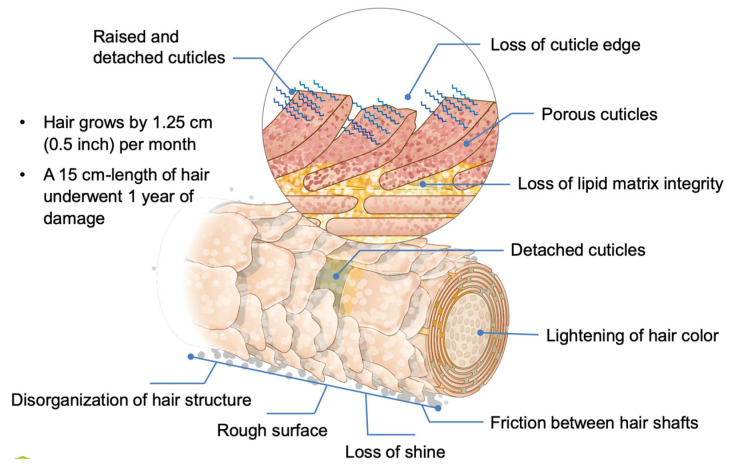
The effects of hair damaging factors [30].

**Figure 11 polymers-14-00749-f011:**
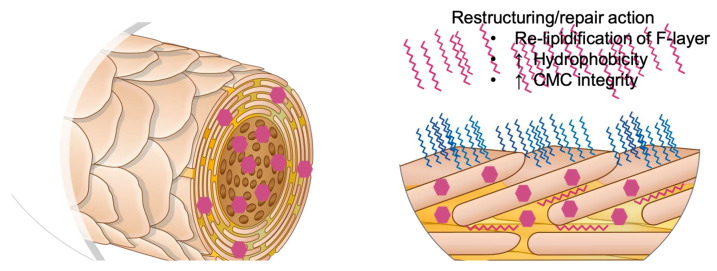
The schematic operation of the discussed product [30].

In the future, the team intends to conduct research on biodegradable polymers derived from waste used in the food industry.


**Techno Economic**


As part of the project, on the basis of research and technological tests carried out in laboratory conditions, it is assumed that optimal technological and technical solutions will be developed, taking into account economic and environmental aspects, and, above all, the health and commercial quality of obtaining products with increased added value. It is also assumed that selected technological operations will be checked on a limited scale on a model test stand and trial batches of new semi-finished products and products with defined pro-health properties will be obtained on it. Based on the results obtained, assumptions for a pilot line for processing fish waste materials will be developed.

## Figures and Tables

**Figure 1 polymers-14-00749-f001:**
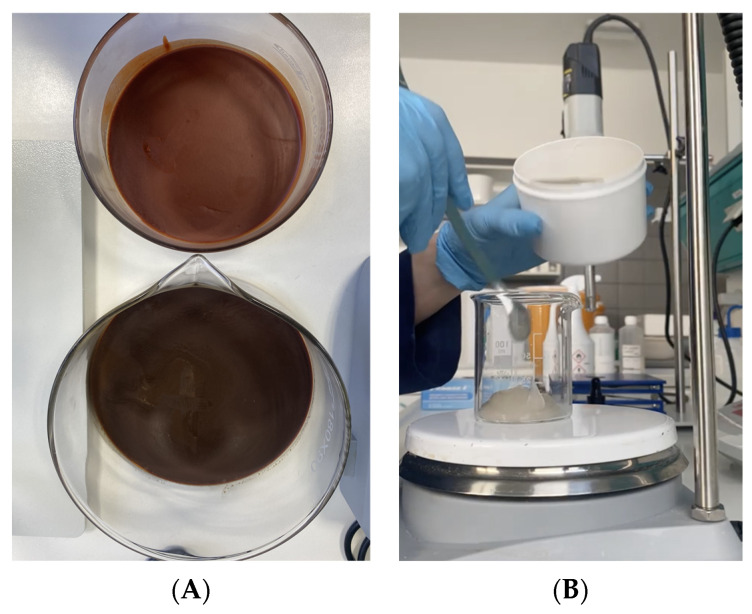
(**A**) Molten phase II. (**B**) The final form of the laminate.

**Figure 2 polymers-14-00749-f002:**
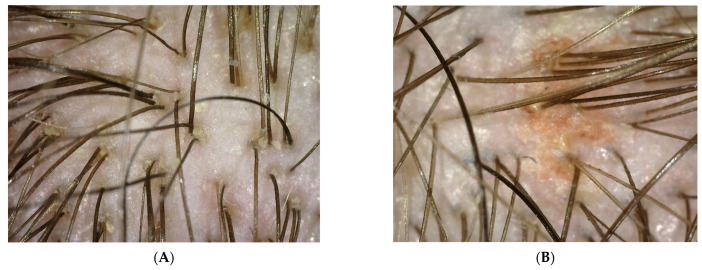
(**A**,**B**) Photo of the scalp from probands before the application of collagen laminate.

**Figure 3 polymers-14-00749-f003:**
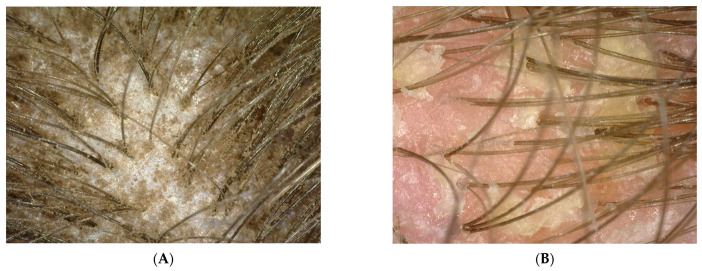
(**A**,**B**) Photo of the scalp of probands V and X, who will continue their care with synthetic organosilicon polymers.

**Figure 4 polymers-14-00749-f004:**
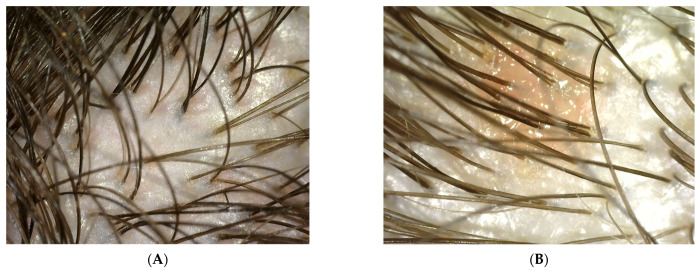
(**A**,**B**) Photo of the scalp from probands after 100 days of collagen laminate application.

**Figure 5 polymers-14-00749-f005:**
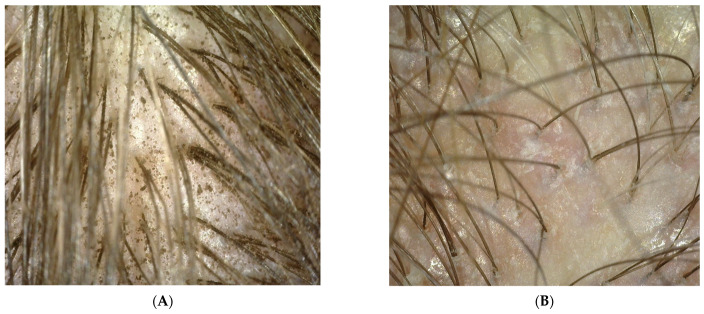
(**A**,**B**) Photo of the scalp of probands V and X after 100 days, who continued the care with synthetic organosilicon polymers.

**Figure 6 polymers-14-00749-f006:**
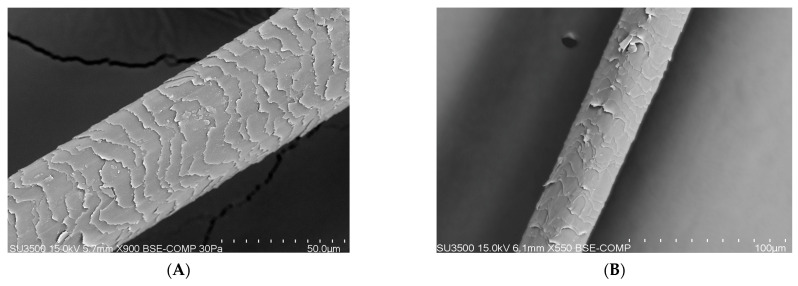
(**A**) Photo of the hair of proband A before the application of collagen laminate (50-micron × 900, 15 kilowatt-hours). (**B**) Photo of the hair of proband B before the application of collagen laminate (100-micron × 550, 15 kilowatt-hours).

**Figure 7 polymers-14-00749-f007:**
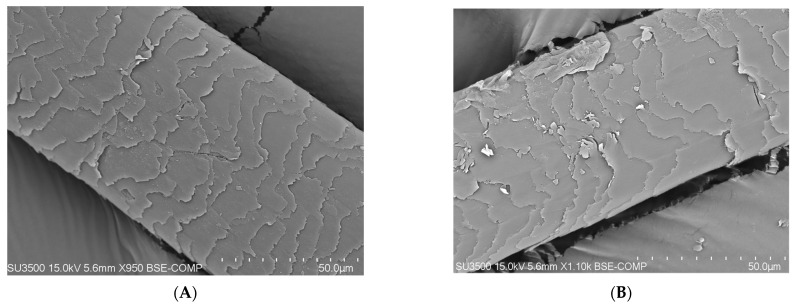
(**A**) Photo of the hair of proband V using care products containing synthetic organosilicon polymers (50-micron × 950, 15 kilowatt-hours). (**B**) Photo of the hair of probant X using care products containing synthetic organosilicon polymers (50-micron × 1000, 15 kilowatt-hours).

**Figure 8 polymers-14-00749-f008:**
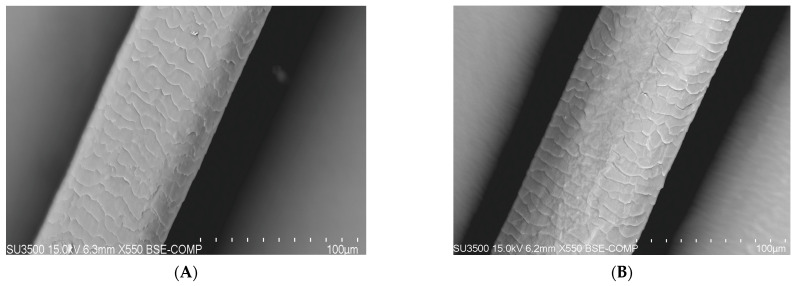
(**A**) Photo of the hair from proband A after 100 days of application of collagen laminate (100-micron × 550, 15 kilowatt-hours). (**B**) Photo of the hair of proband B after 100 days of application of collagen laminate (100-micron × 550, 15 kilowatt-hours).

**Figure 9 polymers-14-00749-f009:**
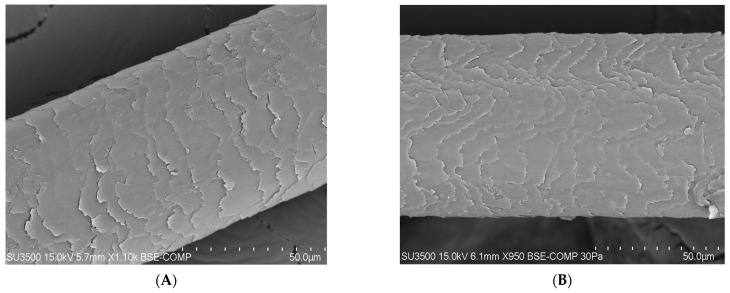
(**A**) Photo of the hair from proband V using care products containing synthetic organosilicon polymers (50-micron × 1000, 15 kilowatt-hour). (**B**) Photo of the hair of proband X using care products containing synthetic organosilicon polymers (50-micron × 950, 15 kilowatt-hour).

**Table 1 polymers-14-00749-t001:** Recipe of collagen laminate.

Trade Name	INCI
Phase I	
Ceramide G and A	Grapeseed/avocado Oil Aminopropanediol Esters
Raspberry	Raspberry Seed Oil/Tocopherol Succinate Aminopropanediol Esters
Phase II	
Collagen	Collagen
Hyaluronic acid	Sodium Hyaluronate
Parfum	Srublet Bukiet

## Data Availability

The data presented in this study are available on request from the corresponding author.

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
