# Peer review of "The Use of Natural Collagen Obtained from Fish Waste in Hair Styling and Care"

_polymers, 2022, doi:10.3390/polym14040749_

Round 1
Reviewer 1 Report
I read an interesting and comprehensive research work entitled ‘The use of biodegradable polymers obtained from fish waste in hair styling and care’. The concept of the article is interesting and suitable to publish in Polymers. This manuscript is generally well written and clearly presented however still needs to address some comments, and thus require substantial major revision.
Title should be modified in the precise way.
Abstract looks very general and not informative should be rewritten. In the abstract authors should mention the importance of research work briefly. Give results values.
In the introduction section, write the novelty of the work and the problem statement clearly. Add more details about the different treatments used for hair treatments and their quantitative data.
The detailed discussion about the novelty, significance of your research work and research gap relative to the literature is essential.
Statistical analysis of the results should be provided in the materials and methods section. It's important for all experimental work Report these values in the results and discussion.
Need to rearrange Results and discussion section. This manuscript lacked substantial discussion of results woithe the recent literature authors should concentrate on this during revision.
For figure and table captions give all details which are quite expected. Don’t use any abbreviation.
Techno Economic challenges of the developed materials need to be addressed. What are the limitations and future research directions that need to be described by adding a new section before the conclusions section?
What are the limitations of using this methodology for commercial application ?.
The conclusion of the study needs to be added with the specific output obtained from the study, it could be modified with precise outcomes with a take home message.
Some English and grammar mistakes are present that need to be correct to improve the quality of the manuscript.
Author Response
Good morning, I sanding my article with amendments.
Cover letter:
1.Title should be modified in the precise way -
Abstract: Chemically speaking, polymers are multi-molecular compounds that have specific physicochemical properties. Hair cosmetics utilize their ability to create a protective film and make the cosmetic formulation more viscous, which facilitates its application. Natural polymers are encountered in nature, but in hair cosmetics artificially modified ones are more often used. Unfortunately, artificially modified polymers are characterized by high resistance to biological factors, which creates an ecological problem. Another reason for a search for natural polymers is their milder action when compared to synthetic ones. One of the new sources of obtaining collagen are waste connective tissue materials of aquatic animals - skins, spines, dorsal chords and scales, and swim bladders. These raw materials are most often disposed of in landfills, processed into fish meal, or destined for food for fur animals. The conducted research was aimed at proving the action of natural collagen in hair cosmetics as a substitute for synthetic polymers. In patients using collagen laminate, it is possible to notice the complete elimination of excessive sebum production, restoration of the correct pH value and reduction of skin inflammations.Based on the assessment of the situation, the condition of the skin and skin can be improved 100%.
3. In the introduction section, write the novelty of the work and the problem statement clearly. Add more details about the different treatments used for hair treatments and their quantitative data.
As already mentioned in the research, the introduction of alternately was undertaken for synthetically obtained polymers, which were used as products for styling and hair care. These compounds were replaced with collagen obtained from fish waste.
Collagen is a building block of skin, bones and connective tissue membranes. It makes up about 30 percent of the protein contained in mammals. Thanks to its unique physicochemical properties, it is widely used in industry, including pharmaceutical and cosmetic.
Fish processing for consumption generates significant amounts of various types of waste materials, which are generally used as an ingredient of animal feed. There are indications of the possibility of a more rational use of some of them for market products intended for humans, and not only for fodder.
Until now, mammalian collagen has been used in the world of cosmetology. However, as a result of the Creutzfeldt-Jacob disease - BSE epidemic. This collagen is systematically replaced with a preparation obtained from fish with much better cosmetic properties, which is related to the use of the so-called acid hydration, which allows the structure of the triple helix to be preserved. This allows for the preservation of the biological activity of collagen, which, by penetrating the extracellular space of the epidermis, influences the activity of keratocytes and fibroblasts, resulting in a number of cosmetic and dermatological effects. Moreover, the solubility of collagen present in the skin of fish allows obtaining the native form of collagen in the final product - collagen hydrogel, using an extremely delicate method of obtaining this protein under conservative conditions [8,9]. The obtained collagen is used both in home hair care, but also as an active substance constituting the basis for professional treatments such as "Botox", lamination and collagen hair straightening. According to statistics, hair care products are the most frequently chosen care products in recent years.
Statistical analysis of the results should be provided in the materials and methods section. It's important for all experimental work Report these values in the results and discussion.
**p<0.01 as a significant improvement from week 0 to week 14 (W14), determined by the post-hoc test. Total trial group (n=30). Measurements were taken every fourteen weeks. For figure and table captions give all details which are quite expected. Don’t use any abbreviation - I changed.Techno Economic challenges of the developed materials need to be addressed. What are the limitations and future research directions that need to be described by adding a new section before the conclusions section?
As part of the project, on the basis of research and technological tests carried out in laboratory conditions, it is assumed that optimal technological and technical solutions will be developed, taking into account economic and environmental aspects, and above all, the health and commercial quality of obtaining products with increased added value. It is also assumed that selected technological operations will be checked on a limited scale on a model test stand and trial batches of new semi-finished products and products with defined pro-health properties will be obtained on it. Based on the results obtained, assumptions for a pilot line for processing fish waste materials will be developed.
What are the limitations of using this methodology for commercial application?.
No limits
The conclusion of the study needs to be added with the specific output obtained from the study, it could be modified with precise outcomes with a take home message.
). The developed method allows to obtain collagen in a native form in the final product. The collagen obtained in this way can be used not only in the form of a gel or solution, but also in a solid form, replacing the bovine collagen used so far. The obtained collagen was used in the form of a gel as a hair care product. Based on the assessment of the situation, the condition of the skin and skin can be improved 100%.
Some English and grammar mistakes are present that need to be correct to improve the quality of the manuscript.
I corrected
Reviewer 2 Report
Abstract needs to be rewritten. A review of currently applied polymers was lacking in the Introduction. The Introduction did not well reflect the innovation and significance of the research.
The author should explain why it is interesting to do the experiments they describe and especially what is the significance of their work.According to these purposes, the present Introduction seems to be prolix and does not reflect the novelty of this article. In my opinion, it may be better to introduce the significance of research from the perspective of sustainable development. Reducing unnecessary emissions and enhancing waste recycling are two important ways to realize the concept of sustainability.The reuse of waste or by-products in agro-food industry (fish waste) can increase economic value and environmental benefits and better highlight sustainability. It might be better to simplify and better explain with realistic examples to evidence the need to reuse agro-food waste by-product. doi: 10.1016/j.lwt.2021.111617, doi:10.3390/foods10081892. The development of biodegradable films should be introduced. For example: doi:10.3390/foods9040449, doi:10.1016/j.fbio.2019.100522
The paper lacked the characterization of the material itself and its properties. Section Conclusions should show the author's conclusions from research rather than citing other people's pictures. It is recommended that the authors redesign the experimental protocol and rewrite the manuscript.
Author Response
Abstract: Chemically speaking, polymers are multi-molecular compounds that have specific physicochemical properties. Hair cosmetics utilize their ability to create a protective film and make the cosmetic formulation more viscous, which facilitates its application. Natural polymers are encountered in nature, but in hair cosmetics artificially modified ones are more often used. Unfortunately, artificially modified polymers are characterized by high resistance to biological factors, which creates an ecological problem. Another reason for a search for natural polymers is their milder action when compared to synthetic ones. One of the new sources of obtaining collagen are waste connective tissue materials of aquatic animals - skins, spines, dorsal chords and scales, and swim bladders. These raw materials are most often disposed of in landfills, processed into fish meal, or destined for food for fur animals. The conducted research was aimed at proving the action of natural collagen in hair cosmetics as a substitute for synthetic polymers. In patients using collagen laminate, it is possible to notice the complete elimination of excessive sebum production, restoration of the correct pH value and reduction of skin inflammations.As already mentioned in the research, the introduction of alternately was undertaken for synthetically obtained polymers, which were used as products for styling and hair care. These compounds were replaced with collagen obtained from fish waste.
Collagen is a building block of skin, bones and connective tissue membranes. It makes up about 30 percent of the protein contained in mammals. Thanks to its unique physicochemical properties, it is widely used in industry, including pharmaceutical and cosmetic.
Fish processing for consumption generates significant amounts of various types of waste materials, which are generally used as an ingredient of animal feed. There are indications of the possibility of a more rational use of some of them for market products intended for humans, and not only for fodder.
Until now, mammalian collagen has been used in the world of cosmetology. However, as a result of the Creutzfeldt-Jacob disease - BSE epidemic. This collagen is systematically replaced with a preparation obtained from fish with much better cosmetic properties, which is related to the use of the so-called acid hydration, which allows the structure of the triple helix to be preserved. This allows for the preservation of the biological activity of collagen, which, by penetrating the extracellular space of the epidermis, influences the activity of keratocytes and fibroblasts, resulting in a number of cosmetic and dermatological effects. Moreover, the solubility of collagen present in the skin of fish allows obtaining the native form of collagen in the final product - collagen hydrogel, using an extremely delicate method of obtaining this protein under conservative conditions [8,9]. The obtained collagen is used both in home hair care, but also as an active substance constituting the basis for professional treatments such as "Botox", lamination and collagen hair straightening. According to statistics, hair care products are the most frequently chosen care products in recent years.
Knowledge of the biochemical structure of collagen, follow the knowledge of the amino acid composition. The collagen triple helix is made of three tropocollagen chains connected with each other by means of covalent-technical bonds by introducing into the middle-glycine. The spiral structure that can be achieved with alanine, alanine, glutamine, histidine, leucine, methionine, tyrosine and tryptophan. Valine and isoleucine due to the large size of the chains, which cannot participate in the formation of the structure of the helix generator. Serine, threonine, proline, and hydroxyproline provide good helix structures. The first two groups from the linkage of hydrogen bonds formed by their hydroxyl ones [17-20]. In the case of proline, the atomic atoms are in a heterocyclic compound, excluding the possibility of rotation around the carbon-nitrogen carbon and formation of hydrogen bonds inside. As a first proposition of proline, the chain may bend or even form a loop.Techno Economic
As part of the project, on the basis of research and technological tests carried out in laboratory conditions, it is assumed that optimal technological and technical solutions will be developed, taking into account economic and environmental aspects, and above all, the health and commercial quality of obtaining products with increased added value. It is also assumed that selected technological operations will be checked on a limited scale on a model test stand and trial batches of new semi-finished products and products with defined pro-health properties will be obtained on it. Based on the results obtained, assumptions for a pilot line for processing fish waste materials will be developed.
Round 2
Reviewer 1 Report
The authors have substantially revised the manuscript according to the comments.
The present form of the manuscript can be accepted for publication.
Author Response
Thank you very much
Reviewer 2 Report
I don't think the author has any valid response to my comment.
Author Response
During the shelf life, it is used for hair care and hair styling on a synthetic basis on silicone.
As already mentioned in the research, it was attempted to introduce alternatives to synthetically obtained polymers that were used as products for styling and hair care. These compounds were replaced with collagen obtained from fish excrements. The research is innovative due to the fact that the described cosmetic product contains natural protein - collagen that builds hair.
Collagen is a building block of skin, bones and connective tissue membranes. It makes up about 30 percent of the protein contained in mammals. Thanks to its unique physicochemical properties, it is widely used in industry, including pharmaceutical and cosmetic.
Fish processing for consumption generates significant amounts of various types of waste materials, which are generally used as an ingredient of animal feed. There are indications of the possibility of a more rational use of some of them for market products intended for humans, and not only for fodder. The obtained collagen has a conditioning and regenerating effect. It fills in cavities in our hair.
Until now, mammalian collagen has been used in the world of cosmetology. However, as a result of the Creutzfeldt-Jacob disease - BSE epidemic. This collagen is systematically replaced with a preparation obtained from fish with much better cosmetic properties, which is related to the use of the so-called acid hydration, which allows the structure of the triple helix to be preserved. This allows for the preservation of the biological activity of collagen, which, by penetrating the extracellular space of the epidermis, influences the activity of keratocytes and fibroblasts, resulting in a number of cosmetic and dermatological effects. Moreover, the solubility of collagen present in the skin of fish allows obtaining the native form of collagen in the final product - collagen hydrogel, using an extremely delicate method of obtaining this protein under conservative conditions [8,9]. The obtained collagen is used both in home hair care, but also as an active substance constituting the basis for professional treatments such as "Botox", lamination and collagen hair straightening. According to statistics, hair care products are the most frequently chosen care products in recent years. The research highlights an important aspect of sustainable development.Reduction of unnecessary emissions and an increase in waste recycling were introduced. These are two important ways to implement the concept of sustainable development. Reusing waste or by-products in the agro-food industry (fish waste) can increase economic value and environmental benefits, and better enhance sustainability. The study used a tactic of combining waste recovery tactics with traditional aquaculture techniques. Cooperation was established with the fish fema, from which the fish residues are not thrown away, and the bacteria are used to produce proteins, which are used to produce the cosmetic in question. The aim of the project was to find a realistic way to recover farm waste. An innovative, integrated aquaculture system has been developed that reduces water consumption and the disposal of waste and nutrients to the environment.
Collagen laminate is a universal product that can be used both for washing, conditioning and styling hair (Figure 9 ). The developed method allows to obtain collagen in a native form in the final product. The collagen obtained in this way can be used not only in the form of a gel or solution, but also in a solid form, replacing the bovine collagen used so far. The obtained collagen was used in the form of a gel as a hair care product. Based on the assessment of the situation, the condition of the skin and skin can be improved 100%.